# An Energy Modulation Interrogation Technique for Monitoring the Adhesive Joint Integrity Using the Full Spectral Response of Fiber Bragg Grating Sensors

**DOI:** 10.3390/s25010036

**Published:** 2024-12-25

**Authors:** Chow-Shing Shin, Tzu-Chieh Lin, Shun-Hsuan Huang

**Affiliations:** Department of Mechanical Engineering, National Taiwan University, No. 1, Sec. 4, Roosevelt Road, Taipei 10617, Taiwan; r04522534@ntu.edu.tw (T.-C.L.);

**Keywords:** adhesive joint, integrity monitoring, fiber Bragg grating, tensile damage, full spectral response, energy modulation interrogation

## Abstract

Adhesive joining has the severe limitation that damages/defects developed in the bondline are difficult to assess. Conventional non-destructive examination (NDE) techniques are adequate to reveal disbonding defects in fabrication and delamination near the end of service life but are not helpful in detecting and monitoring in-service degradation of the joint. Several techniques suitable for long-term joint integrity monitoring are proposed. Fiber Bragg grating (FBG) sensors embedded in the joint are one of the promising candidates. It has the advantages of being close to the damage and immune to environmental attack and electromagnetic interference. Damage and disbonding inside an adhesive joint will give rise to a non-uniform strain field that may bring about peak splitting and chirping of the FBG spectrum. It is shown that the evolution of the full spectral responses can closely reveal the development of damages inside the adhesive joints during tensile and fatigue failures. However, recording and comparing the successive full spectra in the course of damage is tedious and can be subjective. An energy modulation interrogation technique is proposed using a pair of tunable optical filters. Changes in the full FBG spectral responses are modulated by the filters and converted into a conveniently measurable voltage output by photodiodes. Monitoring damage development can then be easily automated, and the technique is well-suited for practical applications. Filter spectrum width of 5 nm and initial overlap with the FBG spectrum to give 40% of the maximum output voltage is found to be optimal for measurement. The technique is tested on embedded FBGs from different adhesive lap-joint specimens and successfully reflected the severity of changes in the full spectral shapes during the course of tensile failure. Moreover, the trends in these PD outputs corroborate with the *V* value previously proposed to describe the qualitative change in FBG spectral shape.

## 1. Introduction

Adhesively bonded joints are superior to conventional rivets, bolts, and weld joints in a number of ways. Load transfer is distributed over a large joining area instead of concentrating on a number of holes with fasteners. This leads to a more advantageous strength-to-weight ratio and stiffness [1,2]. Since the curing temperature of the adhesive is low, and so there will not be material degradation in a heat-affected zone like that in welded joints. The exclusion of stress-concentration holes also helps to avoid fiber discontinuity in composite materials. Thus, adhesive joints are increasingly being employed in aerospace, automotive, and maritime structures. Despite the various advantages of adhesive joints, they cannot be dismantled to allow the materials around the bondline to be inspected for defects/degradation. As in the cases of all practical load-bearing structures, those involving adhesive joining are subjected to various service loading and adverse environments. Such loading may accumulate damages and degrade the joints and, in serious cases, can lead to catastrophic structural failures. For example, the aircraft accident report on the 1988 Aloha Airlines Flight 243 incident traced fuselage failure to a degraded adhesive bond [3].

Non-destructive examination (NDE) techniques are available to reveal defects in adhesive joints. Commonly employed NDE techniques make use of principles involving ultrasonics [4,5,6,7], shearography [8,9], thermography [10,11], and electromechanical impedance [12,13,14]. The application of these NDE techniques is often time- and money-consuming and cannot provide continuous monitoring output. Moreover, most NDE for defects techniques are helpful in detecting disbonding at the fabrication stage and extensive delamination at the late stage of failure. They have difficulty detecting kissing bonds or long-term gradual joint degradation [15].

Instead of directly looking for defects, there are relatively more economical techniques suitable for continuous real-time monitoring of the integrity degradation of adhesive joints. Electrical impedance, strain measuring with strain gauges, and optical fiber sensors are attempted in this category. The electrical impedance method usually requires the adhesive joints to be conductive. This was commonly achieved by mixing carbon nanotubes into the adhesive to form a conductive network. Damages in this network will change the impedance [16,17,18]. Joint damages will change the local strain and thus alter the overall joint stiffness. These changes may be picked up with conventional strain gauges [19,20]. As strain gauge embedded inside the joint may lead to delamination defects, they are usually applied to the external surface and so are subjected to environmental degradation. Additionally, they are susceptible to early fatigue failure. Optical fibers have significantly longer fatigue lives than foil strain gauges [21]. They are compatible with resin adhesive and can be embedded inside the bond without causing internal defects. This will leave the external surface smooth, which is an advantage for structures that need aerodynamic performance. Glass optical fibers are relatively free from environmental attack. As information is modulated in the light signal, fiber sensors are immune to electromagnetic interference and have been commonly employed in structural health monitoring [22,23,24]. Furthermore, multiple sensors can co-exist on the same fiber.

There are two main categories of fiber sensors: distributed [25,26,27] and discrete sensors [28,29,30,31,32,33,34,35,36,37,38,39]. Both are attempted to monitor the integrity of adhesive joints. For distributed sensors, the whole fiber is available for sensing, and it is possible to locate the strain at any specific segment along the fiber. Thus, local perturbation in strain caused by damages in the bond can be revealed. The most commonly employed discrete fiber sensor is the fiber Bragg grating (FBG). An FBG is a series of uniform periodic variations of refractive index in a section of an optical fiber. When a broadband light traveling in the fiber encounters an FBG, a characteristic single-peak spectrum related to the refractive index period will be reflected. The shift of this peak wavelength was often recorded and was related to the strain on the FBG [28,29,30,31,32,33,34]. In this way, FBGs were actually treated as embeddable strain gauges. These FBG strain sensors do possess some advantages over conventional metal foil gauges. Besides the favorable properties mentioned above, embeddability enables the sensors to be closer to the internal damage. However, one must note that peak shifting to reflect strain on the FBG is meaningful when the whole spectrum shifts while preserving its spectral shape. Damages in the adhesive bond and its corresponding strain perturbation are stochastic and localized events. They may result in highly non-uniform strain distribution along an FBG. This fact is reflected in Refs. [35,36,37] where chirp FBG sensors were employed. Chirp FBG has a known varying refractive index period along the grating instead of a single uniform period. A series of wavelengths instead of a single peak will, therefore, be reflected. It is possible to correlate a particular wavelength with the location on the grating, and so chirp FBG offers a certain spatial resolution. By embedding chirp FBG in the adhesive [35] or in one adherend close to the bondline [36,37], it has been shown that a dip in intensity in the full spectral response occurred at locations over artificially induced [35] or naturally initiated disbonds [36,37]. This indicated the occurrence of non-uniform strain over the defects. For single-peak FBG, non-uniform strain distribution will also disrupt the uniform periodicity. This will result in peak splitting and broadening of the FBG spectrum, deforming the original spectral shape. Such changes are schematically shown in Figure 1. As a result, using the shift of a peak in the original spectrum relative to a peak in the deformed spectrum to deduce strain can be problematic. Peak splitting and broadening of embedded FBG spectra in lap joints during tensile and fatigue failures have indeed been observed [38,39,40]. The evolution of the full spectral response of an embedded FBG, instead of a simplistic peak shift, is more indicative of the initiation and development of damages.

Applications making use of the full spectral responses are limited [38,39,40,41]. Webb et al. [38,41] obtain the full spectrum through a post-processing transformation of the time history of intensity recorded with a MEMs-tuned optical filter, whose wavelength varies over time in a controlled manner [42]. Their full spectrum showed heavy broadening and peak splitting. Although the full spectrum was obtained, only one peak was logged, and the FBG essentially functioned as a strain gauge. Our previous works used an optical spectrum analyzer to record the full FBG spectrum [39,40]. The original narrow, single-peaked spectra gradually shifted, broadened, and split into multiple peaks as damages under increasing tensile, fatigue, or hygrothermal damages. After obtaining the spectra at different stages, a comparison of the shapes of successive spectra to discern progressive changes is needed to reveal the occurrence and development of damages. This is not only tedious but also subjective. A parameter has been proposed to allow the change of the spectral shape to be quantified objectively [39,43]. However, the whole process is still time-consuming and inconvenient for practical applications. Additionally, the use of an optical spectrum analyzer to log waveforms is both slow and expensive. In this work, we introduce an energy modulation technique to interrogate the change in the full spectral response. This technique employs two optical filters, and the change in the spectral response is intercepted by the filters and directly output as a voltage in a one-step process. Recording and manual comparison of the FBG spectra are not needed, so this method is much faster and cheaper. It is therefore more convenient and well-suited for practical applications. The width of the filter spectrum and its initial position relative to the FBG spectrum will affect the range and sensitivity of this measurement technique. The effect of these parameters will be examined. Finally, the technique is tested on embedded FBG sensors in a number of composite lap-joint tensile specimens.

## 2. Materials and Methods

### 2.1. Single Lap-Joint Specimens Tensile Testing

Single lap-joint specimens 190 mm in length were fabricated and tested under progressively increasing loading until specimen failures in a servo-hydraulic testing machine (810 Materials Testing System, MTS Systems, Eden Prairie, MN, USA). Three 125 μm diameter FBG-inscribed optical fibers, running along the specimen loading axis, were embedded, both as sensors and as spacers, to control the bond line thickness to ~160 μm. Detailed dimensions and procedures for specimen fabrication, as well as specifications of the FBGs, are reported in Ref. [39].

The FBG spectra under different loadings were recorded. It should be noted that even without damage, tensile loading alone will cause deformation of the joint and change the period of the FBGs. To preclude any effect due to loading, testing was periodically interrupted, and the specimen was unloaded to allow reflected spectra to be measured at 0 N.

### 2.2. Energy Modulation Method for FBG Interrogation

When the spectrum reflected from an FBG embedded in an adhesive joint is passed through a filter, only the overlapping hatched area between the filter and FBG spectra can come through, as illustrated in Figure 2. When this resulting spectrum goes into a photodiode detector, an output voltage proportional to the intensity of the incoming spectrum will be generated. When damage occurs in the adhesive joint, the FBG reflected spectrum may drift, broaden, or chirp. With a fixed filter pass spectrum, the resulting overlapping area will change accordingly and is reflected as a changing voltage.

A schematic arrangement to capture this voltage is shown in Figure 3: The broadband light source (BLS—08001, GIP Technology, New Taipei City, Taiwan) goes into port ① of the circulator (CIR1550PM-FC, Thorlab, Newton, NJ, USA). It will come out at port ② into the FBG, reflected back into port 2, and come out at port ③. This reflected FBG spectrum is passed through a tunable filter (OTF-300-03S3, Santec, Komaki, Aichi, Japan) into a photodiode detector (PDA10CS, Thorlab, Newton, NJ, USA). The latter converts the output coming through the filter into a voltage proportional to the light energy of the overlapping hatched area. The tunable range of the OTF-300-03S3 filter employed is 1530 nm–1590 nm, and the full width at half maximum (FWHM) of the filter spectrum is 1 nm.

### 2.3. Simulation of the Energy Modulation Interrogation Results

As is evident from Figure 2, when the FBG spectrum shifts towards the right, the overlapping area, as well as the voltage output from the photodiode (PD), will increase. However, voltage output will revert to decrease when the FBG spectrum moves on, passing the maximum of the filter spectrum. Thus, during the course of joint damage development, the same voltage output corresponding to two different FBG spectral positions/shapes may arise. A filter spectrum with a larger FWHM helps to alleviate/avoid this situation. However, a larger FWHM implies a more gentle slope of the filter spectrum. This will sacrifice the sensing sensitivity, as a shift in the FBG spectrum against a gentler filter slope will result in a smaller change in the overlapping area. Another factor affecting the sensing range and sensitivity is the initial relative position between the filter and FBG spectra or the amount of overlap between the two.

For optimum sensing range and sensitivity, two questions need to be clarified: (1) How much initial amount of overlap should be set? (2) What FWHM of the filter spectrum should be employed? To answer these questions, instead of trial and error by experimenting with different filters and different initial overlap settings using lap-joint specimens, the calculation of the measured voltages in simulated experiments is attempted to help shed some light on better arrangements.

Three spectra must be known before the simulation calculation can be carried out. The broadband light source (BLS) spectrum was first measured with an optical spectrum analyzer (OSA). Then, the light from the BLS was passed through the filter, and the output was recorded as the filter spectrum. Finally, the FBG-reflected spectrum was measured at the output from port 3 of the circulator in Figure 3.

For any particular wavelength *λ_i_*, the fraction of energy that can come through the filter from a light source, $f_{i}$, is given by the following:(1)$$f_{i}=\frac{the\;light\;intensity\;at\;\lambda_{i}\;of\;the\;filter\;spectrum}{the\;light\;intensity\;at\;\lambda_{i}\;of\;the\;BLS\;spectrum}$$

In the light circuit shown in Figure 3, the output light intensity *I_i_* at *λ_i_* after coming through the filter is as follows:(2)$$I_{i}=f_{i}\times the\;light\;intensity\;at\;\lambda_{i}\;of\;the\;FBG\;spectrum$$

The total energy that the PD receives in Figure 3 is by adding the *I_i_* for every wavelength *λ_i_*. The resulting output voltage *V* is as follows:(3)$$V=C\times \sum_{i}I_{i}$$

*C* is a conversion factor. It was calibrated by passing some known spectra into the PD. The total intensity of this spectrum could be calculated. The output voltage measured by the PD divided by the total intensity calculated gives *C*.

The only filters at hand have a FWHM of ~1 nm. To simulate the filter spectrum with a larger FWHM, we tried to scale up the filter spectrum in the widthwise direction. To facilitate this operation, a curve fitting exercise was first made to the measured filter spectrum. When the center wavelength was set to 1537.157 nm, the OTF-300-03S3 filter spectrum can be fitted with the following quadratic expression:(4)$$y=-147.94{\left(x-1537.157\right)}^{2}-28.845$$
where *y* is the intensity in dBm at wavelength *x* nm. Figure 4 compares the measured OTF-300-03S3 filter spectrum with the above quadratic fit, calculated between 1536.5 nm and 1537.6 nm. Figure 4a shows that above ~−60 dbm, the expression in Equation (4) fit well with the measured spectrum. Much discrepancy seems to exist below −60 dbm. However, the light energy associated with −60 dbm or below is very small, and so such discrepancy has a negligible effect on the subsequent energy modulation output. In fact, if light intensity is expressed in the linear scale in nW, the difference between the quadratic fit and the measured filter spectrum is indeed negligible (see Figure 4b).

To simulate the filter spectrum with a wider FWHM, the expression in Equation (4) was modified to the following:(5)$$y=-147.94{\left(\frac{x-1537.157}{a}\right)}^{2}-28.845$$
where *a* is the intended FWHM in nm. Figure 5a, Figure 5b and Figure 5c display the simulated filter spectra with FWHM equal to 1 nm, 3 nm, and 5 nm, respectively.

Different initial relative positions between the filter and FBG spectra were achieved by tuning the filter center wavelength. This simply shifts the whole filter spectrum and the $f_{i}$’s on the wavelength axis without changing its shape.

## 3. Results

### 3.1. FBG Spectrum Evolution in the Course of Tensile Failure

Figure 6a shows the spectra from an embedded FBG in an adhesive lap-joint specimen under different loads during a typical tensile test. As the applied tensile loading increased from 0 to 3200 N, the spectra progressively shifted towards longer wavelengths. Apart from the appearance of some small secondary peaks, the shape of the spectrum more or less retained its original shape before the test. This shift is caused by the longitudinal extension of the specimen that strains the FBG and increases its period. The emergence of the minor secondary peaks may be attributed to the uneven stress/strain distributions along the longitudinal direction of the joint. Finite element analysis of a single lap joint [44,45] indicated that significant stress concentration occurs right at or very close to the longitudinal edges of the joint for each of the stress components. However, as the stiffness of the optical fiber is much higher than that of the polymeric adhesive, the corresponding strain concentration on the fiber is milder. This may explain the mild secondary peaks at these stages.

At 5200 N, besides shifting further to the right, secondary peaks become more marked. Broadening or chirping became very prominent at 6600 N. This specimen eventually failed at 6797 N. Besides aggravating stress/strain concentration under these high loads, internal damages in the adhesive joint may probably start to emerge and develop at some locations, leading to increasingly profound perturbation of the uniform periodicity, which resulted in the heavy broadening of the spectrum.

In the above spectra, both loading and damage inside the joint played some roles in the strain distribution and contributed to the change in the spectrum shape. As our aim is damage monitoring, the effect of loading on the FBG must be excluded. The latter can be achieved by measuring at zero load. Any deviation of the instantaneous spectrum from the initial damage-free spectrum under the same load will indicate the occurrence and development of damages. To this end, after loading progressively to different levels, the specimens were unloaded for the load-free spectra to be measured.

The evolution of these unload spectra is shown in Figure 6b. The unload spectra from 3200 N or smaller loads virtually overlapped with the spectrum recorded at 0 N at the beginning of the tensile test. Thus, only the unload spectra from 3200 N and above are displayed. An unchanging unload spectrum suggests that negligible damage has arisen at 3200 N. The 5200 N unload spectra exhibited slight deviation from the reference, indicating damage has probably commenced. The 6600 N unload spectra were heavily chirped, showing that damage has become extensive.

### 3.2. Simulation of Energy Modulation Interrogation of FBG Spectrum Development

#### 3.2.1. Verification of the Simulation of Energy Modulation Interrogation Results

To test whether the simulation of the energy modulation interrogation reflects the actual measurement, typical calculated and measured filter-modulated spectra are compared in Figure 7 and Figure 8. Using the circuit shown in Figure 3 and tuning the filter spectrum from short to long wavelengths crossing the FBG spectrum, the maximum output voltage, *V*_max_, was recorded at some point. The relative positions between the filter and FBG spectra were then set, respectively, at PD output of 0.5 *V*_max_ (Figure 7a) and 0.1 *V*_max_ (Figure 8a). Figure 7b and Figure 8b compare the measured output spectra from the filter with the calculated one with the intensity shown in dbm for the respective settings. Above ~−70 dbm, both the measured and the calculated curves agreed well with each other. Marked discrepancies are observed below −70 dbm. The cut-off intensity of the OSA is ~−80 dbm, and the signal intensity below that is displayed as noise. There is no cut-off for the simulation calculation, giving rise to the apparent discrepancies in Figure 7b and Figure 8b. However, as pointed out before, the dbm axes in these two figures are in log scale. The light energy below −60 dbm is negligibly small in absolute value and has no significant impact on the total energy output measured. When expressed in terms of nW, the measured and calculated spectra virtually overlapped with each other, as clearly seen in Figure 7c and Figure 8c.

Next, the measured and calculated PD outputs throughout a tensile test are compared. An FBG embedded inside a single lap-joint specimen was connected to the light circuit in Figure 3. The output from port ③ was fed into a 50–50% coupler to split the light output into two equal halves. Each half was fed into an identical filter. One filter spectrum was set on the short wavelength side to an initial overlap that gave a PD output of 0.5 *V*_max_. The other filter spectrum was set on the long wavelength side to give a PD output of 0.25 *V*_max_. These filter spectra and the FBG spectrum before the test are shown in Figure 9a. The specimen was then pulled under tension at incrementally increasing load. After each load increment, the specimen was unloaded. The FBG spectrum and the voltage outputs from the two filters were recorded. Some typical load-free FBG spectra are shown in Figure 9b. These spectra are not needed in actual measurement using the proposed energy modulation technique. These recorded FBG spectra were used for the calculation of the simulated voltage outputs. The latter are compared with the outputs measured from the PDs, and the results are shown in Figure 10 for filters at the short and long wavelength sides, respectively. As the unloaded FBG spectra in this tensile test remained unchanged before 4000 N, both PD outputs changed little initially. Beyond 4400 N, the change in the FBG spectral shape became more marked, and the spectrum showed a tendency to shift towards the shorter wavelength (see Figure 9b). This direction of shift will move the FBG spectrum into the filter on the left while moving out of the filter on the right. As a result, the output from the filter on the short wavelength side increased to a maximum, and that from the filter on the long wavelength side decreased.

Two observations may be noted. Firstly, the simulated values agree reasonably well with the output measured from the PDs, showing that using the simulation calculation to estimate the PD output is feasible. Secondly, the output from the PD may increase or decrease, depending on the relative positions of the filter and FBG spectra, as well as the direction of shift of the FBG spectra.

Observation from a number of tensile failure monitoring using FBG shows that there is no rule of thumb on whether the spectrum will shift towards the short or long wavelength sides. This is especially true when damage becomes extensive, leading to broadening and splitting the spectrum into multiple peaks. Broadening will not spread the spectrum symmetrically. In some specimens, it may spread more toward the shorter wavelengths, while in the other specimens, it spreads more towards the longer wavelengths. The reason behind this may be explained by the fact that the exact pattern of damage differs in each individual specimen. If a single filter is used, the output voltage may either increase or decrease as damage develops. This ambiguous phenomenon is not desirable when one uses the energy modulation method to judge the extent of damage development. In order to achieve a more intuitive interpretation where the output increases with damage development, a normalized PD output is proposed, making use of a two-filter arrangement with spectra tuned to overlap the FBG spectrum on the short and long wavelength sides, respectively, as follows:(6)$$Normalized\;PD\;Voltage=(\frac{{|V}_{L}-V_{L0}|+|V_{R}-V_{R0}|}{V_{max}})\times 100$$

$V_{L}$ and $V_{R}$ are, respectively, the PD voltage outputs from the filters on the short and long wavelength sides. $V_{L0}$ and $V_{R0}$ are the initial $V_{L}$ and $V_{R}$ before testing. $V_{max}$ is the maximum output voltage recorded when the filter spectrum is tuned from short to long wavelengths crossing the FBG spectrum.

The two-filter arrangement can be set up using the circuit schematically shown in Figure 3 with a slight modification. A 50-50 coupler is inserted at port ③ of the circulator to split the output into two equal halves. To each half, a tunable filter and a photodiode are connected. The filters are first tuned to locate the maximum voltage output *V*_max_. One filter is then tuned down towards the shorter wavelength while the other is tuned up to the longer wavelength to achieve the desired overlap output voltage. These two voltage outputs can then be fed into Equation (6) to get the normalized PD.

#### 3.2.2. Effect of Filter Spectrum Positions Relative to the FBG Spectrum

From the schematic diagram in Figure 2, if the initial overlap between the two spectra is minimal and the FBG spectrum is moving away from the filter spectrum during the course of damage, the overlapping area will quickly drop to zero, and no output will be recorded even though the damage is continuously developing. On the other hand, if initially the spectra fully overlap, a slight shift/change in the FBG spectrum may not change the overlapping area, and early damage development will not be revealed. In order to evaluate how much initial overlap between the filter and FBG spectra will be optimum for damage monitoring, simulation calculation was employed to test a series of different initial overlaps using the load-free spectra from the test shown in Figure 6b. The two-filter arrangement was used. The maximum output voltage $V_{max}$ was first identified before testing. The short- and long-wavelength filters were then tuned away from the maximum overlap position symmetrically against the fixed FBG spectrum. Nine sets of initial overlap configurations were tested, with output in increments of 0.1 $V_{max}$ from 0.1 to 0.9 $V_{max}$. The relative positions between the spectra for 0.1 $V_{max}$ and 0.9 $V_{max}$ are shown, respectively, in Figure 11a,b.

The simulated outputs for four of the nine filter configurations using the unloaded FBG spectra in the course of tensile testing are shown in Figure 12. Figure 12 also includes the *V* value proposed in Ref. [39]. Ref. [39] showed that the damage inside an adhesive joint can qualitatively be reflected in the change of the embedded FBG spectral shape. The *V* value was intended to quantify the change of FBG spectral shape with respect to the reference spectrum before testing to aid objective and quantitative assessment of the development of internal damage. *V* value in Ref. [39] is defined as follows:(7)$$V=\frac{1000\sqrt{\sum_{i=1}^{n}{\left(P_{\lambda_{i}}^{current}-P_{\lambda_{i}}^{reference}\right)}^{2}}}{\sum_{i=1}^{n}P_{\lambda_{i}}^{reference}}$$
where *P_λi_^current^* and *P_λi_^reference^* are, respectively, the power intensity in dbm of the current unload spectra and the reference spectra at the same corresponding wavelength *λ_i_*. *n* is the number of data points in the FBG spectrum wavelength span.

Figure 6b shows that the unload FBG spectra from about ~50% failure load have shifted by a small but discernible amount toward a longer wavelength. Beyond ~70% failure load, the unload spectra showed more significant changes. The evolution of the *V* value agrees with these trends. When the initial overlap amounts to 0.1 $V_{max}$, the normalized PD output has nearly no change up to ~70% failure load. The insensitivity of the overlap setting is understandable. With only a small initial overlap, the FBG spectrum has to shift a considerable amount to allow any change to be intercepted by the filter spectrum and reflected in the photodiode output. The sensitivity is significantly improved, and the PD output follows the *V* value closely when the initial overlap amounts to 0.3 $V_{max}$ (Figure 12b). However, when the overlap increases to 0.5 $V_{max}$ or more, the discrepancy with the *V* value occurs close to the failure load where the PD voltage drops despite the obvious increase in damage. An initial overlap to give a PD voltage of 0.3 to 0.4 $V_{max}$ is optimum to reflect the amount of change in the FBG spectra and the damage inside the lap joint. Figure 6b shows that the FBG spectrum has already broadened to slightly larger than 3 nm close to failure. This is well beyond the 1 nm FWHM filter spectrum and can explain the observed drop in PD output close to failure.

#### 3.2.3. Effect of the FWHM of the Filter Spectrum

It is pointed out in Section 2.3 that another factor affecting the sensitivity of the energy modulation method is the width of the filter spectrum. The filter employed for measurement has a spectrum FWHM of 1 nm. The actual effect of filter spectrum FWHM is examined by simulation for FWHM up to 5 nm. The results for FWHM of 3 nm and 5 nm for initial overlap give a PD voltage of 0.4 and 0.9 $V_{max}$ are shown in Figure 13 to illustrate the trends.

For the 3 nm FWHM filter with an initial overlap of 0.4 *V*_max_, the PD output agreed well with the *V* value before ~50% of the failure load (Figure 13a). Beyond this point, the *V* value started to increase at a greater rate because of a more marked change in the FBG spectrum. The PD output failed to reflect this accelerated rate, indicating that the sensitivity of a 3 nm FWHM filter is somewhat inferior to the 1 nm filter. On the other hand, unlike that of the 1 nm filter, increasing the initial overlap from 0.4 *V*_max_ through to 0.9 *V*_max_ does not significantly improve the sensitivity, as is evident in Figure 13b. Figure 13b also shows that with a wider filter spectrum, increasing the initial overlap does not bring about a large abnormal drop in PD output near failure. Figure 13c,d shows that the 5 nm FWHM filter exhibited similar characteristics.

For the same peak intensity, a filter spectrum with a wider FWHM will have a gentler slope and, thus, a lower sensitivity towards change in the FBG spectrum. However, a filter spectrum with a wider FWHM does not miss out on a large broadening and shifting of the FBG spectrum. For general applications, the shape of the FBG spectrum varies, and some may have chirped and broadened considerably after embedding into a structure. Additionally, large deformation in a structure will cause considerable wavelength shifts in the FBG spectrum. Figure 6b shows that near failure, the single-peaked FBG spectrum has broadened to slightly beyond 3 nm. To be on the safe side, a filter with a wide FWHM is more advisable.

### 3.3. Testing the Technique on More Embedded FBG in Lap-Joint Tensile Specimens

To further verify the feasibility of this method for monitoring the integrity of lap joints, we tried to apply it to monitor more embedded FBG in different lap-joint tensile tests. The results in Section 3.2 recommend a filter spectrum FWHM of 5 nm and an amount of overlap to give a PD output of 0.4 *V*_max_. As the filter at hand only has an FWHM of 1 nm, we again resort to simulation calculation to obtain the results for 5 nm FWHM. Figure 14a,c,e are spectrum evolutions from 3 FBGs. For clarity, only a few typical spectra are drawn in these figures. The bracketed numbers following the loading values in the figure legends indicated the loading as a percentage of the final failure load. Figure 14b,d,f are the corresponding PD voltages using the energy modulation technique and the *V* values. All three FBG spectra changed a little up to 50–60% of the respective failure loading of the lap-joint specimens. This is reflected in small and gradual increases in both the PD voltages and the V values in Figure 14b,d,f. The FBG3 spectra in Figure 14a started to show serious broadening and peak splitting after loading reached ~80% of the failure load. Correspondingly, Figure 14b shows the PD voltage started to increase steeply at this point. FBG4 also started to show broadening after reaching ~80% of the failure load, but the amount of broadening was gentler (see Figure 14c) when compared with that in FBG3. The corresponding acceleration in PD output increase is also gentler, and the final voltage reaching close to failure is also much smaller (see Figure 14d). FBG 5 showed only very moderate broadening even close to failure (see Figure 14e), and the corresponding PD output also increased gradually without marked acceleration in change (see Figure 14f). Appendix A provides further data for another 3 FBGs, and the general development in the monitored outputs matched with the development of the full FBG spectra. In all three cases, the trend in the *V* values corroborated closely with that in the PD output voltage, suggesting both parameters correctly reflected the extent of shape changes in the evolution of the full FBG spectra during progressive tensile failure. The major difference between the two is that the *V* value requires an optical spectrum analyzer to log the full FBG spectrum followed by computation afterward, while the PD voltage can be obtained simply using suitable optical filters and photodiodes, and no follow-up computation is needed. Thus, the proposed energy modulation technique is relatively cheaper, much faster, easily automated, and more convenient for interrogating the development of the full FBG spectrum.

## 4. Conclusions

Previous work has demonstrated that the load-induced damages inside an adhesive lap joint are reflected as spectrum shifting, peak splitting, the emergence of secondary peaks, and broadening of the spectra of embedded fiber Bragg gratings. To interrogate the full FBG spectral responses more conveniently, an energy modulation technique is proposed and examined. This technique employs two optical filters with pass spectra, respectively overlapping with the short and long wavelength sides of the FBG spectrum. The pass-through intensity from each filter is then converted to a voltage output by a photodiode. The voltage outputs from both filters are then combined to a normalized PD output voltage. Filter spectrum FWHM of 5 nm and initial overlap of 0.4 *V*_max_ is recommended, and the technique is tested on embedded FBGs from different adhesive lap-joint specimens and successfully reflected the severity of changes in the full spectral shapes during the course of tensile failure. Moreover, the trends in these PD outputs corroborate with the *V* value previously proposed to describe the qualitative change in FBG spectral shape. The proposed technique is a one-step process doing away with the slow and expensive optical spectrum analyzer and the recording and manual comparison of the FBG spectra. Thus, it is fast, cheap, convenient, and well-suited for practical applications.

## Figures and Tables

**Figure 1 sensors-25-00036-f001:**
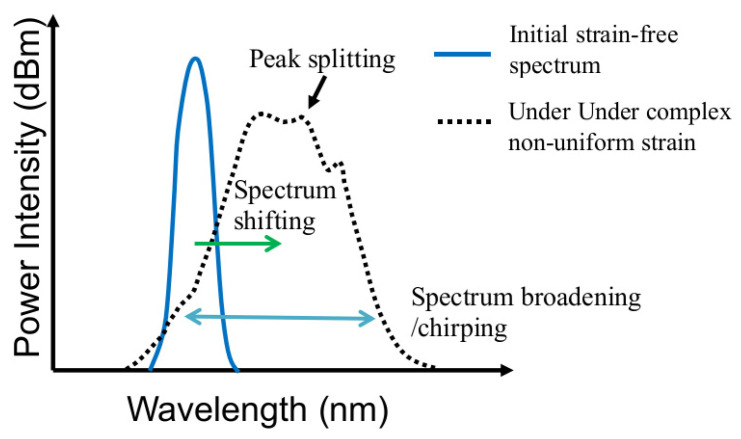
Schematic diagram showing the effect of complex strain field on the spectral shape reflected from an FBG.

**Figure 2 sensors-25-00036-f002:**
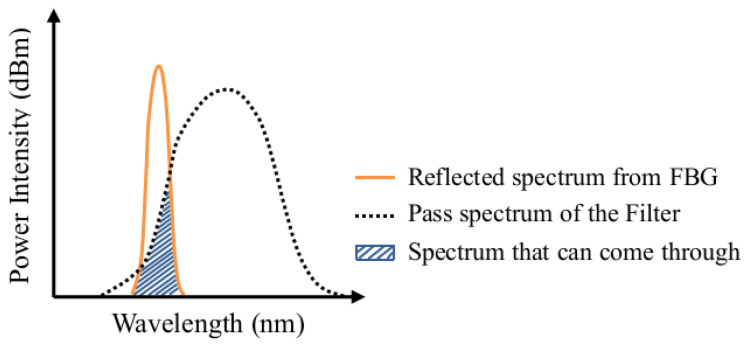
Schematic arrangement for energy modulation interrogation of FBG.

**Figure 3 sensors-25-00036-f003:**
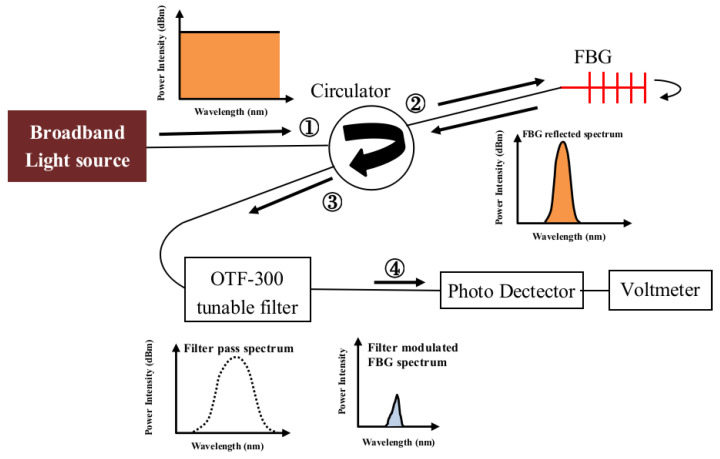
Overlapping of the FBG and filter spectra resulted in the hatched area that will come through the filter.

**Figure 4 sensors-25-00036-f004:**
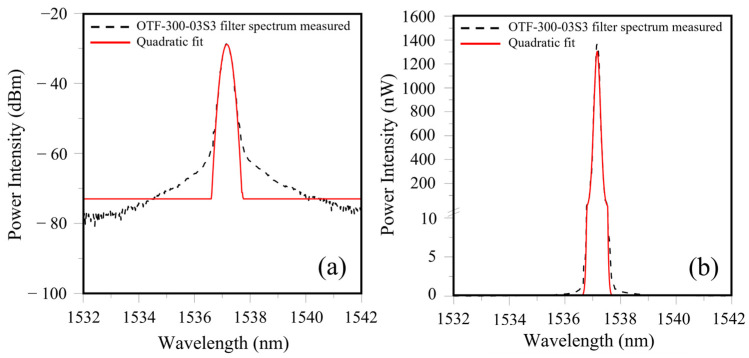
Comparison of measured filter spectrum with the quadratic fit with light intensity in (**a**) dbm; and (**b**) nW.

**Figure 5 sensors-25-00036-f005:**
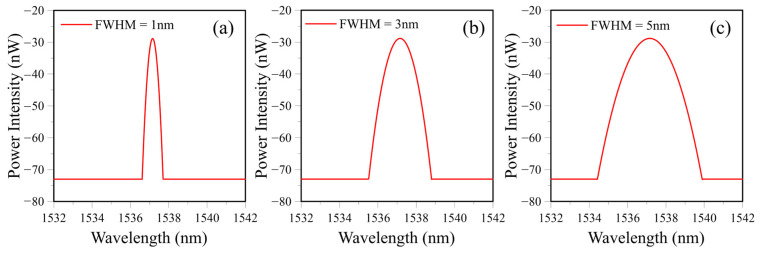
Simulated filter spectra for full width at half maximum equals (**a**) 1 nm, (**b**) 3 nm, and (**c**) 5 nm.

**Figure 6 sensors-25-00036-f006:**
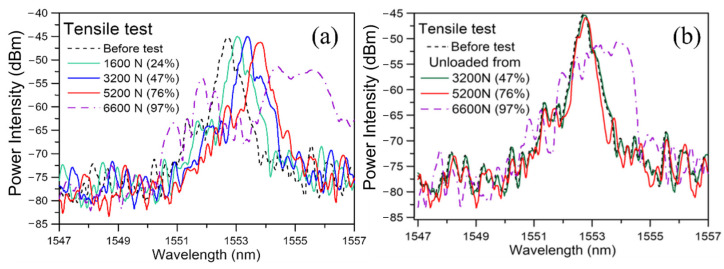
The evolution of FBG spectrum measured at (**a**) increasing load during a tensile test; (**b**) zero load after progressively increasing load during a tensile test (the bracketed numbers after the loading values in the figure legends indicated the loading as a percentage of the failure load).

**Figure 7 sensors-25-00036-f007:**
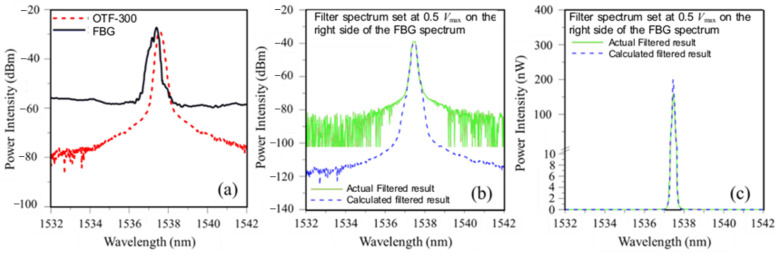
Comparison of calculated and measured energy-modulated results: (**a**) Relative positions between the filter and FBG spectra to give 0.5 of maximum voltage; (**b**) energy-modulated output spectra in dbm; (**c**) energy-modulated output spectra in nW.

**Figure 8 sensors-25-00036-f008:**
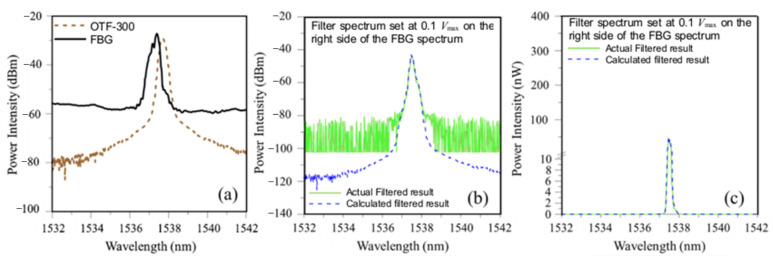
Comparison of calculated and measured energy-modulated results: (**a**) Relative positions between the filter and FBG spectra to give 0.1 of maximum voltage; (**b**) energy-modulated output spectra in dbm; (**c**) energy-modulated output spectra in nW.

**Figure 9 sensors-25-00036-f009:**
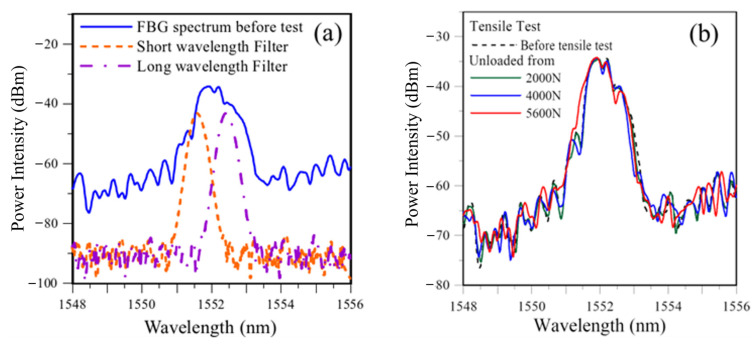
(**a**) The FBG and the two-filter spectra before the test; (**b**) evolution of the unloaded FBG spectra up to 96% of failure load.

**Figure 10 sensors-25-00036-f010:**
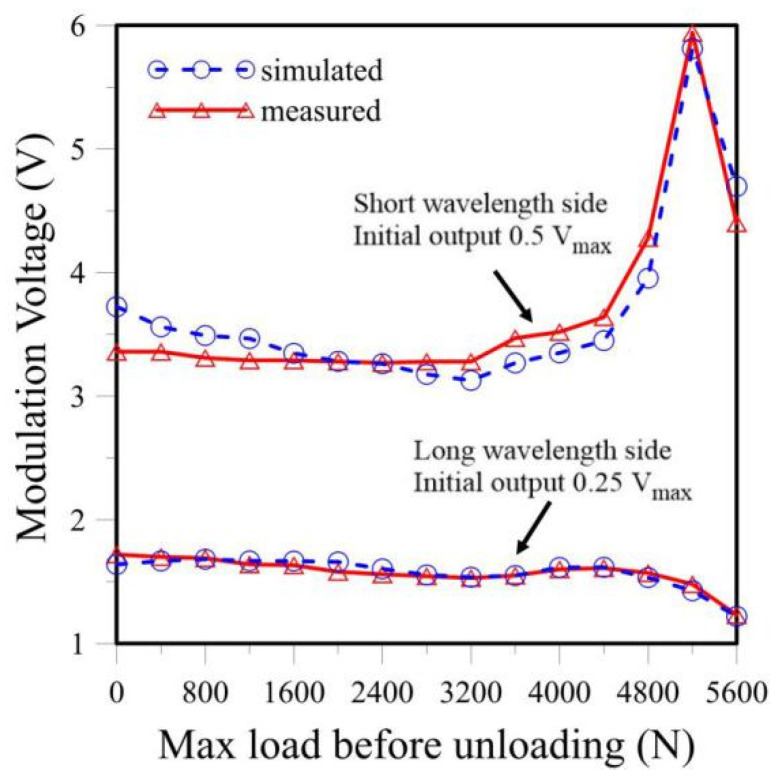
Comparison between the measured and simulated energy modulation outputs from the filter on the short wavelength side, with an initial output of 0.5 *V*_max,_ and on the long wavelength side, with an initial output of 0.25 *V*_max_.

**Figure 11 sensors-25-00036-f011:**
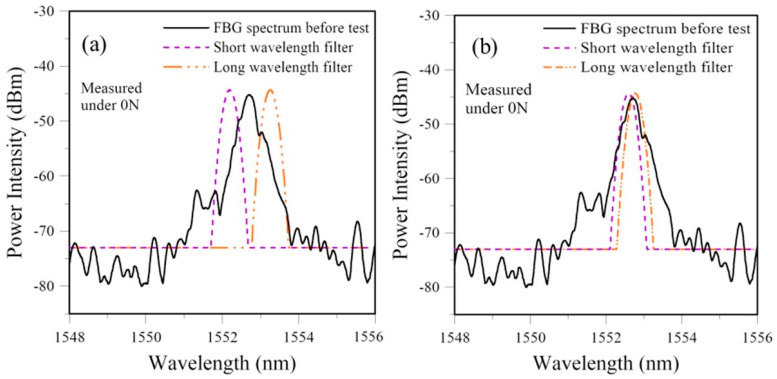
Relative positions between the filters and FBG spectra when output equals (**a**) 0.1 $V_{max}$ and (**b**) 0.9 $V_{max}$.

**Figure 12 sensors-25-00036-f012:**
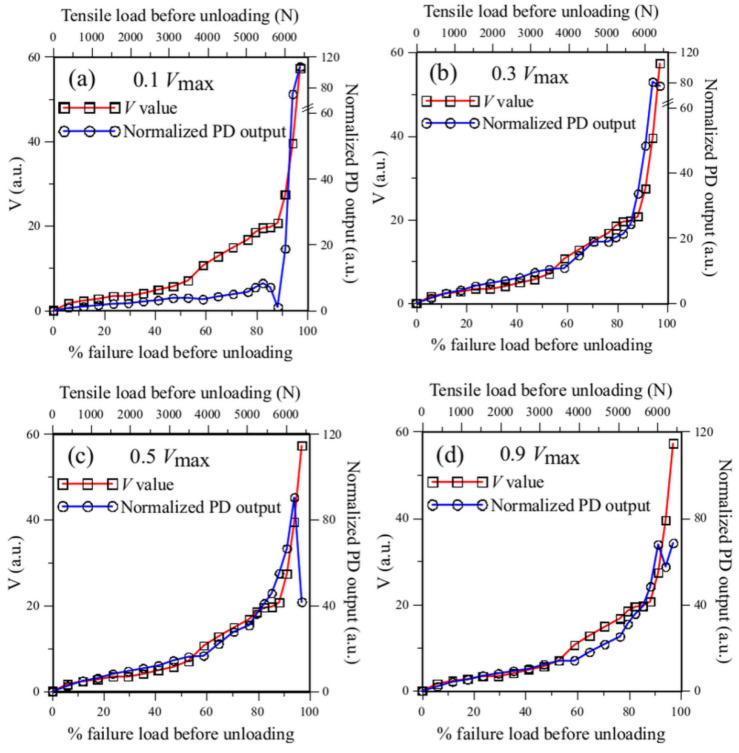
Simulated PD output compared with the *V* values for initial overlap corresponding to (**a**) 0.1 $V_{max}$; (**b**) 0.3 $V_{max}$; (**c**) 0.5 $V_{max}$; and (**d**) 0.9 $V_{max}$.

**Figure 13 sensors-25-00036-f013:**
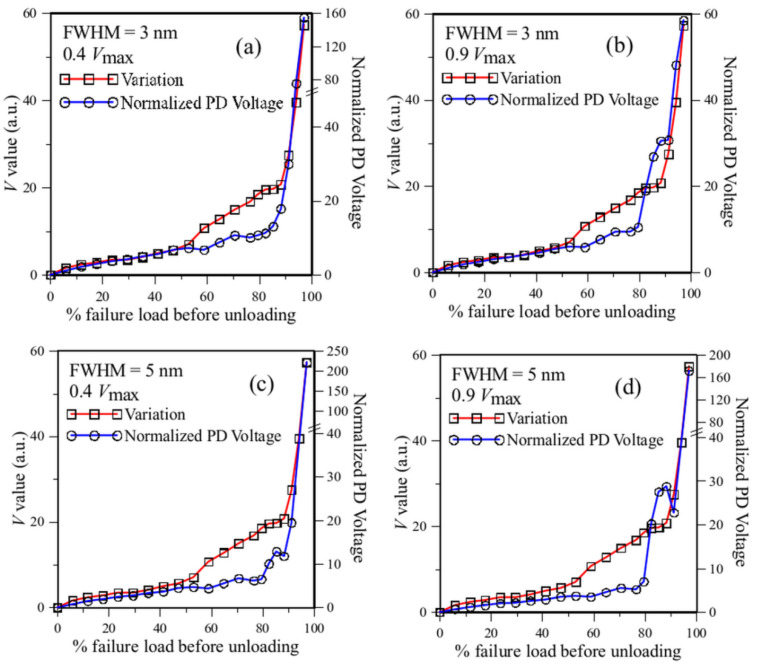
Simulated PD output for filters with different FWHMs and initial overlap PD voltages compared with the *V* values: (**a**) FWHM = 3 nm, 0.4 $V_{max}$; (**b**) FWHM = 3 nm, 0.9 $V_{max}$; (**c**) FWHM = 5 nm, 0.4 $V_{max}$; and (**d**) FWHM = 5 nm, 0.9 $V_{max}$.

**Figure 14 sensors-25-00036-f014:**
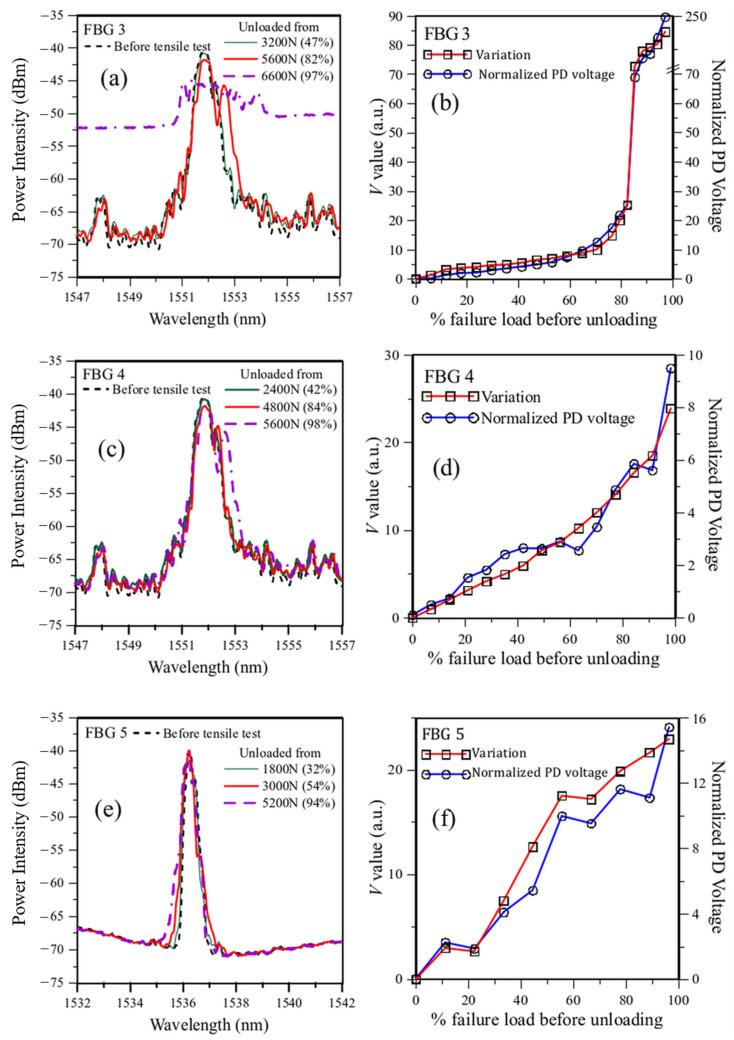
Evolution of the load-free spectra during tensile tests in (**a**) FBG 3; (**c**) FBG 4; and (**e**) FBG 5; and comparison of the corresponding V values and simulated PD output for filters. FWHM = 5 nm and initial overlap of 0.4 $V_{max}$ for (**b**) FBG 3; (**d**) FBG 4, and (**f**) FBG 5. (the bracketed numbers after the loading values in the figure legends indicated the loading as a percentage of the failure load).

## Data Availability

Data are contained with the article and Appendix A.

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
