# Peer review of "An Energy Modulation Interrogation Technique for Monitoring the Adhesive Joint Integrity Using the Full Spectral Response of Fiber Bragg Grating Sensors"

_sensors, 2024, doi:10.3390/s25010036_

Round 1
Reviewer 1 Report
Comments and Suggestions for Authors
Manuscript entitled “An Energy Modulation Interrogation Technique for Monitoring the Adhesive Joint Integrity Using the Full Spectral Response of Fiber Bragg Grating Sensors”, by Chow-Shing Shin, et al., introduced an energy modulation technique to interrogate the change in the full spectral response. This method employs two optical filters, and the change in the spectral response is intercepted by the filters and directly output as a voltage. The results showed in the manuscript are interesting, but there are still some problems:
1. In the abstract sections, some results should be added. Meanwhile, the content of abstract can be appropriately shortened.
2. In introduction section, the author needs to introduce the relevant reports in detail, and demonstrate the research significance and purpose of this work.
3. Overall discussion is ok however it should be improved, especially the mechanism of the Simulation of energy modulation interrogation of FBG spectrum development. In addition, the author needs further condense the innovation of the paper.
4. In the manuscript, too many Figures suggest merging, such as Figure 6 and 7, 10 and 11, etc.
5. Overall, this manuscript is less practical, and the author should use more data to support the conclusion.
Author Response
We have made changes according to each point of the comments and suggestions. Please refer to the attached file for details.

Reviewer 2 Report
Comments and Suggestions for Authors
I believe that the results presented in this manuscript are of undoubted interest and deserve publication in the presented form. The developed fiber-optic sensor of stresses in adhesive joints is easy to operate, does not require expensive equipment and gives reliable results. The authors presented a sufficient number of experimental results confirming the possibility of unambiguous interpretation of signals from only two photodetectors in terms of the load experienced by the adhesive joint.
Author Response
Thank you very much for the comment and we totally agree with it. In fact, it is the aim of the paper to provide the relevant communities with a convenient and low cost method for monitoring the structural health degradation of adhesive joints.
Reviewer 3 Report
Comments and Suggestions for Authors
Type of manuscript: Article
Title: An Energy Modulation Interrogation Technique for Monitoring the
Adhesive Joint Integrity Using the Full Spectral Response of Fiber Bragg Grating Sensors
Journal: Sensors
Manuscript ID: sensors-335339
Authors: Chow-Shing Shin *, Tzu-Chieh Lin, Shun-Hsuan Huang
Recommendation: Major Revisions.
The paper presents a study of Fiber Bragg Grating Sensors. Fiber Bragg grating sensors can measure a physical phenomenon that can be translated into deformation. Thus, they are universal sensors that can be used to evaluate a wide variety of physical phenomena. These sensors can be integrated into the structure being measured during the production stage. This will allow monitoring the degradation of the joint in real time. Thus, the properties of these sensors are of practical interest and are currently being actively studied. The key role in using these sensors is played by signal processing and interpretation. The authors performed a detailed study of the properties of the fiber Bragg grating sensors. The authors proposed an energy modulation method. This study demonstrates the effective use of sensors to monitor damage in adhesive joints. In the process of studying this article, several questions arose that the authors need to answer.
1. Does the sensor degrade the adhesive joint?
2. Does IR radiation affect the adhesive joint? Will constant IR monitoring degrade the adhesive joint properties?
3. Is there repeatability? If we examine another object, will the scheme proposed by the authors work? Or do we need to recalibrate?
4. Will it be cost-effective to use such a sensor instead of standard sensors?
5. Why did the authors use the IR spectrum? Why didn't they use coherent laser light?
Author Response
We have responded to each point of the comments. Please refer to the attached file for details.

Round 2
Reviewer 1 Report
Comments and Suggestions for Authors
The revised manuscript has addressed the issues proposed by the reviewer. Now it can be accepted in this current version.
Comments on the Quality of English LanguageThe quality of English language is ok.
Reviewer 3 Report
Comments and Suggestions for Authors
The authors have answered all questions. Accept.